## [Peer Review File · Nature Communications]

Observation of site-selective chemical bond changes via ultrafast chemical shiftsEditorial Note: This manuscript has been previously reviewed at another journal that is not operating a transparent peer review scheme. This document only contains reviewer comments and rebuttal letters for versions considered at *Nature Communications*.

REVIEWER COMMENTS

Reviewer #2 (Remarks to the Author):

I would like to thank the authors for responding to the comments. I recommend publication of the manuscript in *Nature Communications*. I nevertheless think, that a couple of edits, outlined below, will be useful.

The authors now cite previous time-resolved core-photoelectron studies saying ‘Recent studies [7, 8] have partially shown the potential of this technique to follow UV-excited state dynamics, but a higher temporal resolution was desired in order to track the transient evolving system via chemical shifts right after excitation [9]. In this work we extend the concept’

I do not think that the authors extend the concept to a level that would be applicable to UV-excited states, as UV pulse length as well as wavepacket dispersion would anyhow make the methods used in this paper invalid in the UV case. There is no doubt that they do something great in the domain of x-ray excitation and x-ray probe. The value of their work speaks for itself, even without the argument for making UV –experiments better.

The authors now clarify, that they subtract the 40 fs spectrum from all experimental spectra in Fig. 2.

I think it should be written in the caption and the y-axel label should be ‘difference amplitude’ instead of amplitude. These changes would make the point clear immediately.

My original question regarding Fig. 3: d) ‘Why is the wavepacket already at early times bimodal? It would also be good to mention the vibrational splitting and its relation to the Auger lifetime.’

The authors added a new figure in supplementary information, that is valuable. From that it seems, that a 1 fs pulse does not create a ‘bimodal’ wavepacket. I think this is known in vibrational wavepacket theory. I would suggest that the authors simply mention the vibrational period in the state. The 10 fs is apparently longer than that, which structures the wavepacket to resemble more a vibrational eigenstate with the close to bimodal behavior.

Reviewer #3 (Remarks to the Author):

I thank the authors for their reply to my original review. After reading the reply, all the reviews and their respective replies by the authors, and studying the revised manuscript, I now concur that, while pump-probe experiments have been at the core of ultrafast science for quite some time, being able to add element-specific chemical information to the pump, provides a critical new capability that I had not fully appreciated. To make this aspect of providing a core-level pump clearer, the authors should consider changing the title to something like 'Observation of element-selective chemical bond changes via ultrafast chemical shifts'. In the current title (as well as the original title), it is not clear that the pump is also element/site selective, and not just the probe.

With regards to the lack of data quality, I do appreciate the challenges in such a proof-of-principle experiment, and I believe that the authors have addressed my and the other reviewers' concerns. Improvements in x-ray free-electron laser pulse control and diagnostics will improve the data quality in future experiments. The readers can see for themselves, how well the data is supported by the theory provided by the authors, and what challenges lie ahead before this x-ray pump-x-ray probe approach will become a useful tool for tracking the ultrafast electron rearrangement and chemical bond breaking site-specifically and in real time.

I recommend publication.

Reviewer #4 (Remarks to the Author):

The manuscript by A. Al-Haddad, et.al. concerns an experimental measurement and theoretical calculations examining an x-ray pump/x-ray probe experiment at LCLS to monitor the effects of time delaying the pulses in a two-color experiment where the pump promotes a 1s electron from the oxygen atom of carbon monoxide to the π^* state and the probe follows the core-excited intermediate into Auger decay pathways (bound and dissociative). The authors find a noticeable chemical shift in the carbon 1s environment which they attribute to the electronic dynamics that evolve

from the pumped state. This study represents an ambitious effort to resolve the complicated evolutionary study of core-excited molecular states and pushes the state of the art in terms

of x-ray pump/ x-ray probe experiments as well as the theoretical

capabilities to describe these processes and monitor the ultrafast electron rearrangement. The experiment and theory reported are novel in the exploration of the evolving electronic environment that gives rise to the chemical shift (measured as the central peak photoelectron energy moderated from the usual bonding energy of the carbon K-edge) and do seem appropriate for consideration to advance ultrafast studies.

However, the quality of the agreement between experimental

measurements and the accompanying theory calculations do seem to leave lingering questions that don't necessarily present a complete picture supporting the conclusions of the work. In particular, comparison of these results presented in Figure 2(a) and (b) show variability in the largest intensity of the core-excited states with differences in the order of the time-delayed results (max for theory with zero delay and notably larger in the experiment at 5fs, also calculated to be broader in energy and with significant differences for the longest time delays that theory predicts are essentially flat. The relative magnitudes within panels 2(a) and (b) also seems very different

between experiment vs theory. Further, the results for the transient signals vs time delay in panel (c) shows better agreement between the two, but with experiment having a broader max peak and a shoulder through 15 and 20 fs, where the results in (a) and (b) have different trends at 20fs. In the lower panels examining the Auger states, better agreement is present in predicting the order of the dominant peak order for each time delay presented, but the secondary peak order doesn't really share that agreement. The location of the peaks in the

experiments do seem to exhibit more variability in the binding energy than the theory which displays more magnitude variation rather than broadening for the longest time delays.

I am curious why the energy scales on the horizontal axis for these results are not consistently presented in the lower panels of Fig. 2, as they are in the top row panels 2(a) and 2(b); this would make direct comparison easier to read. In examining the integrated results for the transient signal in panel (f) the authors state that the measured and simulated transient signals are "in good agreement", but some significant differences are not (and need to be) discussed, such as the slower rise of theory into longer time delays compared to experiment for the 3.5eV range at the highest energy window considered (red) and why the blue window results in the experiment return to increasing after the peak (that best agrees between theory and experiment) above 10fs. Some explanation of the discrepancy is absolutely necessary to have confidence in which of the measurements

or the theory is more accurate.

The issue of coherence effects are better addressed in the Supplementary Material (SM), where the authors provide a table of

the populated Auger states with their decay widths, showing the

separation of energy that would be observed in the ejected electron being fairly well separated and distinguished from each state. That seems appropriate in that the overlap of kinetic energies would be rather distinct (unlike the satellite states themselves).

The partial decay widths are incoherently summed (over J in Eq. 10 in SM) and govern the dynamics as a square root of

this sum. This will incorporate the magnitude of these overlaps, but not their phases, and perhaps should be briefly justified, as well. Also, Eq. 14 of the SM argues that the electronic amplitude of the photoelectron yield can be approximated as a product of factors (the authors call them "two terms" in the SM, but "two factors" is more appropriate): (1) a dipole coupling term between the $1s$ of the target and the continuum photoelectron, which at the large kinetic energies that result ($>200\text{eV}$) would lead to rather flat (unvarying) amplitudes/cross sections and (2) an overlap of electronic wave functions that suddenly shakes up the hole state from the initial, and would more likely be the varying factor responsible for the dynamics. The dismissal of coherence effects in this part relies on their tabulated energy separation.

Overall, it seems clear that the statistics of the experiment are limited, doubtlessly due to the difficulty of collecting data in the SASE pulse variations that require careful examination per shot. At the same time, the theory for calculating the correlated quantum electron and nuclear dynamics and treating the Auger decays phonologically from their experimentally determined widths (since it is challenging to calculate the continuum wave functions for the Auger final states) relies on several approximations that do seem justified, but don't particularly help explain the discrepancies and differences with the experimental data. In summary, there is enough uncertainty and ambiguity in both the theory and experiment themselves to not know which is correct where they don't agree with each

other. With these rather significant discrepancies, it is problematic to publish with questions and lingering doubt of which may be right.

Reply to the Reviewers

Remarks to the authors in blue

Reply to the reviewers in black

I. REVIEWER 2

I would like to thank the authors for responding to the comments. I recommend publication of the manuscript in Nature Communications. I nevertheless think, that a couple of edits, outlined below, will be useful.

We thank the reviewer for his/her support and the suggested edits that we included in our manuscript.

The authors now cite previous time-resolved core-photoelectron studies saying ‘Recent studies [7, 8] have partially shown the potential of this technique to follow UV-excited state dynamics, but a higher temporal resolution was desired in order to track the transient evolving system via chemical shifts right after excitation [9]. In this work we extend the concept’ I do not think that the authors extend the concept to a level that would be applicable to UV-excited states, as UV pulse length as well as wavepacket dispersion would anyhow make the methods used in this paper invalid in the UV case. There is no doubt that they do something great in the domain of x-ray excitation and x-ray probe. The value of their work speaks for itself, even without the argument for making UV –experiments better.

We intended to embed our work within the broader and recent literature but we agree that the context was misleading. In the new version of the manuscript, we kept the same references and adjusted the sentence to clarify our new contribution. The sentence now reads ”In this work we apply the concept of chemical shifts to the few femtosecond regime and extend it with a site-selective trigger. We approach the natural timescale of electron motion and show the sensitivity of chemical shifts to excited electronic state dynamics, electron rearrangement and the correlated nuclear motion.”.

The authors now clarify, that they subtract the 40 fs spectrum from all experimental spectra in Fig. 2. I think it should be written in the caption and the y-axel label should be ‘difference amplitude’ instead of amplitude. These changes would make the point clear immediately.

We modified Fig. 2 and the caption accordingly.

My original question regarding Fig. 3: d) ‘Why is the wavepacket already at early times bimodal? It would also be good to mention the vibrational splitting and its relation to the Auger lifetime.’

The authors added a new figure in supplementary information, that is valuable. From that it seems, that a 1 fs pulse does not create a ‘bimodal’ wavepacket. I think this is known in vibrational wavepacket theory. I would suggest that the authors simply mention the vibrational period in the state. The 10 fs is apparently longer than that, which structures the wavepacket to resemble more a vibrational eigenstate with the close to bimodal behavior.

The core-excited state lifetime is 4.2 fs and the vibrational splitting is ~ 115 meV (average over the energy splitting for the 3 most populated vibrational states), which corresponds to a vibrational period of ~ 36 fs. We added this information in section S2.2 of the SM.

We thank again the reviewer for his/her last suggestions and for the time and effort he/she dedicated to review our manuscript. We hope that with the last edits it is now fully acceptable for publication.

II. REVIEWER 3

I thank the authors for their reply to my original review. After reading the reply, all the reviews and their respective replies by the authors, and studying the revised manuscript, I now concur that, while pump-probe experiments have been at the core of ultrafast science for quite some time, being able add element-specific chemical information to the pump, provides a critical new capability that I had not fully appreciated. To make this aspect of providing a core-level pump clearer, the authors should consider changing the title to something like ‘Observation of element-selective chemical bond changes via ultrafast chemical shifts’. In the current title (as well as the original title), it is not clear that the pump is also element/site selective, and not just the probe.

We thank the reviewer for reconsidering our manuscript and support.

We also appreciate the proposal for the very nice new title which we used as suggested.

With regards to the lack of data quality, I do appreciate the challenges in such a proof-of-principle experiment, and I believe that the authors have addressed my and the other reviewers’ concerns. Improvements in x-ray free-electron laser pulse control and diagnostics will improve the data quality in future experiments. The readers can see for themselves, how well the data is supported the theory provided by the authors, and what challenges lie ahead before

51 this x-ray pump-x-ray probe approach will become a useful tool for tracking the ultrafast electron rearrangement and
 52 chemical bond breaking site-specifically and in real time.

53 I recommend publication.

54 We thank the reviewer for supporting our manuscript for publication, and the appreciation of the quality of our
 55 study taking into consideration the XFEL capabilities at the time of the experiment. Since our original experiments,
 56 the capabilities of XFELs to offer multi-color x-ray pulses have greatly improved with promise for narrower spectral
 57 bandwidth, larger time and spectral separation, and most crucially higher repetition rate. We are therefore optimistic
 58 that the data quality of future studies will greatly improve and trigger many further applications.

59 III. REVIEWER 4

60 The manuscript by A. Al-Haddad, et.al. concerns an experimental measurement and theoretical calculations exam-
 61 ining an x-ray pump/x-ray probe experiment at LCLS to monitor the effects of time delaying the pulses in a two-color
 62 experiment where the pump promotes a 1s electron from the oxygen atom of carbon monoxide to the π^* state and the
 63 probe follows the core-excited intermediate into Auger decay pathways (bound and dissociative). The authors find a
 64 noticeable chemical shift in the carbon 1s environment which they attribute to the electronic dynamics that evolve
 65 from the pumped state. This study represents an ambitious effort to resolve the complicated evolutionary study of
 66 core-excited molecular states and pushes the state of the art in terms of x-ray pump/ x-ray probe experiments as
 67 well as the theoretical capabilities to describe these processes and monitor the ultrafast electron rearrangement. The
 68 experiment and theory reported are novel in the exploration of the evolving electronic environment that gives rise to
 69 the chemical shift (measured as the central peak photoelectron energy moderated from the usual bonding energy of
 70 the carbon K-edge) and do seem appropriate for consideration to advance ultrafast studies.

71 We appreciate the summary and positive comments of the reviewer as well as his/her support of our work. We are
 72 also optimistic that this work will pave the way for many more applications in the ultrafast community by studying
 73 the evolving chemical environment.

74 However, the quality of the agreement between experimental measurements and the accompanying theory calcula-
 75 tions do seem to leave lingering questions that don't necessarily present a complete picture supporting the conclusions
 76 of the work. In particular, comparison of these results presented in Figure 2(a) and (b) show variability in the largest
 77 intensity of the core-excited states with differences in the order of the time-delayed results (max for theory with
 78 zero delay and notably larger in the experiment at 5fs, also calculated to be broader in energy and with significant
 79 differences for the longest time delays that theory predicts are essentially flat. The relative magnitudes within panels
 80 2(a) and (b) also seems very different between experiment vs theory. Further, the results for the transient signals vs
 81 time delay in panel (c) shows better agreement between the two, but with experiment having a broader max peak and
 82 a shoulder through 15 and 20 fs, where the results in (a) and (b) have different trends at 20fs. In the lower panels
 83 examining the Auger states, better agreement is present in predicting the order of the dominant peak order for each
 84 time delay presented, but the secondary peak order doesn't really share that agreement. The location of the peaks
 85 in the experiments do seem to exhibit more variability in the binding energy than the theory which displays more
 86 magnitude variation rather than broadening for the longest time delays.

87 There are several reasons why Figure 2(a) and (b) are not exactly identical. The main reason is that the theoretical
 88 model is not accounting for SASE pulses, but instead assumes transformed-limited pulses. From an experimental
 89 point of view, the core-excited state spectral regime is still dominated by the ground state photo line due to the broad
 90 SASE pulses. Extracting such weak signals from the fluctuating SASE XFEL was challenging as addressed in the SI.
 91 The transient behavior of the core-excited state is also susceptible to timing jitter and pulse shape variability of the
 92 FEL pulses. This limits the temporal resolution and adds uncertainty to locating the absolute time zero by several
 93 femtoseconds. Specific challenges for the Auger energy windows included low count rates and accordingly a poorer
 94 signal to noise ratio. We report on the details about the signal strength and statistics in detail in the SI material.

95 We agree with the referee that the presentation of our results was not ideal and can leave "lingering questions" about
 96 the agreement of experiment and theory, in particular as the information was distributed between the main manuscript
 97 and SI. In response to the criticism we have rewritten large parts of the section II and III. We now specifically point
 98 to the limitations in the experimental data analysis and accuracy of the theoretical predictions right at this point in
 99 the manuscript instead of deferring to the SM. We believe that these improvements of the manuscript makes it more
 100 accessible and alleviate remaining uncertainties.

101 I am curious why the energy scales on the horizontal axis for these results are not consistently presented in the
 102 lower panels of Fig. 2, as they are in the top row panels 2(a) and 2(b); this would make direct comparison easier
 103 to read. In examining the integrated results for the transient signal in panel (f) the authors stat that the measured
 104 and simulated transient signals are "in good agreement", but some significant differences are not (and need to be)
 105 discussed, such as the slower rise of theory into longer time delays compared to experiment for the 3.5eV range at

the highest energy window considered (red) and why the blue window results in the experiment return to increasing after the peak (that best agrees between theory and experiment) above 10fs. Some explanation of the discrepancy is absolutely necessary to have confidence in which of the measurements or the theory is more accurate.

The energy difference between the two main peaks Fig. 2(d) and (e) is indeed smaller in the measurements by 3 eV. An error of this magnitude can be expected considering the level of available electronic structure theory to describe the highly excited Auger and core-hole states in order to calculate the time-resolved XPS spectra. Electron relaxation plays an important role in the final core-hole states, as these states are highly unstable and the electrons rearrange in the presence of a core vacancy to stabilize the system. We have extended the discussion in the first paragraph of Section III by adding:

”Such a level of discrepancy is expected due to the complexity of describing strong electron correlation in core-hole states. In particular, we find here that at long internuclear distances, (more than 50) highly excited core-hole (satellite) states are populated, see Fig. S6 in SM. We can provide an accurate description of the strong electron correlation for the 10 lowest-excited states by using state-average calculations at CASSCF level. However, this approach is not computationally feasible for the highly excited satellites states, which are thus treated at the configuration interaction level without further relaxation of the orbitals. An more exhaustive discussion is provided in Section S1.1.4 of the SM.”

This explains the small discrepancy in the chemical shift (the energy gap between the low and high binding energy peaks). We would like to reiterate that the error in energy is within the expected range of 1% for excited states calculation. This small difference in energy is also the reason for the slightly different x axes in panels d) and e) of Fig. 2. Panel d) shows the experimental energy window and panel e) highlights the same features on the calculated energy axes.

The issue of coherence effects are better addressed in the Supplementary Material (SM), where the authors provide a table of the populated Auger states with their decay widths, showing the separation of energy that would be observed in the ejected electron being fairly well separated and distinguished from each state. That seems appropriate in that the overlap of kinetic energies would be rather distinct (unlike the satellite states themselves).

We appreciate the reviewer comment/support in this point.

The partial decay widths are incoherently summed (over J in Eq. 10 in SM) and govern the dynamics as a square root of this sum. This will incorporate the magnitude of these overlaps, but not their phases, and perhaps should be briefly justified, as well. Also, Eq. 14 of the SM argues that the electronic amplitude of the photoelectron yield can be approximated as a product of factors (the authors call them ”two terms” in the SM, but ”two factors” is more appropriate): (1) a dipole coupling term between the $1s$ of the target and the continuum photoelectron, which at the large kinetic energies that result ($\approx 200\text{eV}$) would lead to rather flat (unvarying) amplitudes/cross sections and (2) an overlap of electronic wave functions that suddenly shakes up the hole state from the initial, and would more likely be the varying factor responsible for the dynamics. The dismissal of coherence effects in this part relies on their tabulated energy separation.

In order to evaluate the total Auger decay width (Eq. (11) in SM) we use the measurements of the partial decay widths published in Ref. [7]. In these type of experiments, the energy of the Auger electron is measured, but not the angular distribution. In our experiment we do not detect the angular distribution of the Auger electrons either, which justifies why in our model we do an incoherent sum over the angular momentum J (see Eq. (10) in SM).

We have added an explanation in the SM in the caption of Table S1 and at the end of Section S1.1.1. that reads as: ’In Eqs. (10) and (11), we do an incoherent sum over the angular momentum. We neglect the phase of the states in Eq. (10) as the Auger states are separated in energy (see Table S1) and thus do not lead to interference during the propagation. In the experiment, we are not measuring the angular momentum distribution such that this approximation is justified in our model.’.

As the referee also suggested, we have added a note after Eq. (14): ’In the present case, coherences do not play a significant role and we treat all the different ionization channels coming from the different Auger paths independently, as they are quite separated in energy, see table S1. Hence, Auger electrons from different channels will be emitted to the continuum with different energies due to energy conservation, with a broadening given by the Auger decay lifetime (0.158 eV), having no overlap among them. This explains why we can separate the ionization processes in Eq. (14).’. We also change in the SM in the explanation of Eq. (14) ”terms” to ”factors”.

Overall, it seems clear that the statistics of the experiment are limited, doubtlessly due to the difficulty of collecting data in the SASE pulse variations that require careful examination per shot. At the same time, the theory for calculating the correlated quantum electron and nuclear dynamics and treating the Auger decays phonologically from their experimentally determined widths (since it is challenging to calculate the continuum wave functions for the Auger final states) relies on several approximations that do seem justified, but don’t particularly help explain the discrepancies and differences with the experimental data. In summary, there is enough uncertainty and ambiguity in both the theory and experiment themselves to not know which is correct where they don’t agree with each other. With these rather significant discrepancies, it is problematic to publish with questions and lingering doubt of which

¹⁶⁴ may be right.

¹⁶⁵ We thank again the reviewer for the comments and thoughts he/she put into the paper. We believe that the
¹⁶⁶ adjustments we made to the manuscript following his/her comments improved the overall paper and clarified the
¹⁶⁷ ambiguities. We hope that also reviewer 4 supports publication of this manuscript in Nature Communications.

REVIEWERS' COMMENTS

Reviewer #4 (Remarks to the Author):

The authors have responded to the criticisms raised in my original referee report and the manuscript addresses the points raised. I am satisfied with the amended draft and would recommend its acceptance for publication.